# Comparing the Effect of Folic Acid and Pentoxifylline on Delaying Dialysis Initiation in Patients with Advanced Chronic Kidney Disease

**DOI:** 10.3390/nu11092192

**Published:** 2019-09-12

**Authors:** Hsun Yang, Shiun-Yang Juang, Kuan-Fu Liao, Yi-Hsin Chen

**Affiliations:** 1Department of Nephrology, Department of Internal Medicine, Taichung Tzu Chi Hospital, Buddhist Tzu Chi Medical Foundation, Taichung 427, Taiwan; yishin0819@gmail.com; 2Department of Medical Research, Taichung Tzu Chi Hospital, Buddhist Tzu Chi Medical Foundation, Taichung 427, Taiwan; rastlinvmp6@yahoo.com.tw; 3Division of Gastroenterology, Department of Internal Medicine, Taichung Tzu Chi Hospital, Buddhist Tzu Chi Medical Foundation, Taichung 427, Taiwan; kuanfuliaog@gmail.com; 4School of Medicine, Tzu Chi University, Hualien 970, Taiwan; 5Graduate Institute of Integrated Medicine, China Medical University, Taichung 404, Taiwan; 6Institute of Clinical Medicine, National Yang-Ming University School of Medicine, Taipei 112, Taiwan

**Keywords:** angiotensin-converting enzyme inhibitor, angiotensin receptor blocker, chronic kidney disease, delay dialysis, erythropoiesis-stimulating agent, folic acid, pentoxifylline

## Abstract

Background: We hypothesized that the nutrient loss and chronic inflammation status may stimulate progression in advanced chronic kidney disease. Therefore, we aimed to generate a study to state the influence of combined nutritional and anti-inflammatory interventions. Methods: The registry from the National Health Insurance Research Database in Taiwan was searched for 20–90 years individuals who had certified end-stage renal disease. From January 2005 through December 2010, the diagnosis code ICD-9 585 (chronic kidney disease, CKD) plus erythropoiesis-stimulating agent (ESA) use was defined as entering advanced chronic kidney disease. The ESA starting date was defined as the first index date, whereas the initiation day of maintenance dialysis was defined as the second index date. The duration between the index dates was analyzed in different medical treatments. Results: There were 10,954 patients analyzed. The combination therapy resulted in the longest duration (*n* = 2184, median 145 days, *p* < 0.001) before the dialysis initiation compared with folic acid (*n* = 5073, median 111 days), pentoxifylline (*n* = 1119, median 102 days, *p* = 0.654), and no drug group (control, *n* = 2578, median 89 days, *p* < 0.001). Lacking eGFR data and the retrospective nature are important limitations. Conclusions: In patients with advanced CKD on the ESA treatment, the combination of folic acid and pentoxifylline was associated with delayed initiation of hemodialysis.

## 1. Introduction

According to the United States Renal Data System (USRDS) 2015 report (2013 registration data), the incidence of end-stage renal disease (ESRD) in Taiwan is still the highest in the world (458 per million population), which is far higher than that of the United States (363 per million population), Japan (286 per million population), and the European countries. At the same time, the prevalence (3138 people per million population) also ranks number one in the world [1]. Considering the huge expense burden of chronic kidney disease (CKD) and the number of patients on dialysis in Taiwan, reducing the incidence of ESRD and identifying the factors that can delay dialysis is a critical issue for nephrologists. The Taiwan health care authorities want to rid the island of the stigma associated with “Dialysis Island.” Therefore, adjuvant therapies that can slow renal function decline in patients with CKD are critically needed.

While a single common etiology has not been identified in the complex process of CKD [2], we hypothesized that nutrient loss because of diet restriction and chronic inflammation contributed by CKD itself may stimulate progression in advanced chronic kidney disease. We tried to review previous large studies about this field and compared the differences between them. Folic acid was then selected as a nutrient intervention because Taiwan was in a folic acid non-fortification area. In the mean time, pentoxifylline was well studied in this field for its anti-inflammatory effects, we also conducted it as a good model for comparison.

To date, several randomized controlled trials of folic acid have focused on assessing the impact of lowering of homocysteine levels on cardiovascular outcomes in patients with CKD [3,4,5]. Although the Hoorn and Hisayama studies have shown that hyperhomocysteinemia is associated with the risk of developing albuminuria and CKD [6,7] and the CoLaus study has further disclosed hyperhomocysteinemia as a risk factor for the progression of CKD [8,9], the effects of lowering homocysteine by folic acid fortification in some randomized controlled trials have shown no benefit or harmful effects on renal outcomes [10,11]. Recently, the investigators of the Renal Substudy of the China Stroke Primary Prevention Trial (CSPPT) have claimed that folic acid can significantly delay the progression of CKD among patients with mild-to-moderate CKD [12], which may position the possible role of folic acid therapy in delaying dialysis and give us a direction to investigate.

It is well known that the combination of an angiotensin-converting enzyme inhibitor (ACEI) or angiotensin receptor blocker (ARB) and pentoxifylline have protective effects in reducing proteinuria by ameliorating the decline in the estimated glomerular filtration rate (eGFR) in patients with stages 3–5 CKD [13,14,15]. However, a systematic review and meta-analysis showed that there was no conclusive evidence proving the usefulness of pentoxifylline for improving renal outcomes in subjects with CKD of various etiologies [16]. The Current Kidney Disease Improving Global Outcomes (KDIGO) clinical practice guideline has noted folic acid and pentoxifylline as erythropoiesis-stimulating agents (ESA) adjuvant therapy for anemia. However, due to lacking of sufficient evidence and convincing data, the guideline also suggests not using the adjuvant ESA treatment for anemia, including vitamin C, vitamin D, vitamin E, folic acid, L-carnitine, and pentoxifylline [17]. Few studies evaluate and compare the efficacy of folic acid and pentoxifylline for delaying the initiation of dialysis.

In this study, we aimed to generate a national wide study in Taiwan to state the influence of combined nutritional and anti-inflammatory interventions to provide potential treatment options, which could delay the progression of advanced chronic kidney disease. Folic acid and pentoxifylline were chosen as our target and the study was designed as an objective observation study. We investigated and compared the effects of folic acid and pentoxifylline on delaying dialysis initiation in predialysis patients with advanced CKD.

## 2. Materials and Methods

### 2.1. Data Source

This study analyzed data obtained from the National Health Insurance Research Database (NHIRD). The data from the National Health Insurance (NHI) program was sorted into files, including registration files and original claims data for reimbursement. These data files were de-identified by scrambling the identification codes of both patients and medical facilities. In the NHI program, almost 95% of the hospitals’ healthcare data and 99% of the entire population of 23 million in Taiwan were enrolled. International Classification of Diseases-9th revision (ICD-9) codes were used to define the diseases investigated. Data for diagnostic codes, medication prescriptions, and medical procedures were also included in the database. In addition, illnesses including congenital illness, cancer, ESRD, and others, were established into a file of “The registry for catastrophic illness patients”.

The data were extracted from the patient files in the Registry for Catastrophic Illness, and all patients were confirmed to have been in a chronic dialysis status for at least three months. Additionally, the absolute and relative indications for initiation of maintenance dialysis are well defined by NHI regulations. The absolute indication is defined as an eGFR <5 mL/min/1.73 m^2^ or serum creatinine (SCr) ≥10 mg/dL. The relative indication contained two parts: (1) Patients with diabetes: eGFR ≤15 mL/min/1.73 m^2^ or SCr ≥6.0 mg/dL and accompanying complications; (2) Patients without diabetes: eGFR ≤10 mL/min/1.73 m^2^ or SCr >8.0 mg/dL and accompanying complications. To avoid increasing the incidence of rickets among patients and causing them to enter the fifth stage of CKD where uremia has already affected the whole body’s organ systems and or to prevent rising mortality, we treated patients with absolute renal failure or relative severe chronic renal failure with dialysis. At least two nephrologists must validate a patient’s illness certification for ESRD based on the patient’s medical records, laboratory examinations, and imaging studies [18].

### 2.2. Study Design and Population

The study design was a retrospective population-based observational study that aimed to investigate and compare the effects of folic acid and pentoxifylline on delaying dialysis initiation in predialysis patients with advanced CKD.

Individuals who had the certification of ESRD within 20–90 years old were enrolled in our study. A primary diagnosis of chronic kidney disease (ICD-9 codes: 585) plus the first day of receiving ESA treatments between 1 January 2005 and 31 December 2010 were identified as the first index date, which indicated the starting of advanced stage 5 CKD. The date these patients started receiving regular hemodialysis of three or more month’s duration was defined as the second index date. The period of the first index date to the second index date was calculated as the predialysis duration in the enrolled patients with advanced stage 5 CKD being treated with ESA. Additionally, the time between the first index date and second index date was analyzed in different treatment groups. The exclusion criteria were: (1) Dialysis initiation before the ESA treatment date; (2) renal transplantation and peritoneal dialysis; (3) no angiotensin-converting enzyme inhibitor or angiotensin receptor blocker (ACEI/ARB) treatment within 90 days after the first index date; (4) death within 90 days after the first index date; (5) cancer before the first index date; (6) missing data.

The method of selecting advanced chronic kidney disease patients (CKD stage 5) from the NHIRD database was carried out from several publications [19,20,21]. According to NHI reimbursement regulations, patients with CKD stage 5 whose serum creatinine >6 mg/dL (approximately equivalent to a glomerular filtration rate <15 mL/min/1.73 m^2^) and a hematocrit <28% should receive ESA to maintain a target hematocrit level not to exceed 36%. Since the Bureau of NHI regularly audits the insurance claims, the orders of the NHI-reimbursed ESA prescribed by nephrologists are usually accurate. Furthermore, an internal report from the Taiwan Department of Health disclosed that the rate of ESA use was 85% in 2012 for patients with advanced stage 5 CKD who had not yet commenced dialysis [22]. Based on Hwang et al. [23], the median level of SCr at dialysis initiation in Taiwan is 10.1 mg/dL, corresponding to a median eGFR level of 4.7 mL/min/1.73 m^2^, and the patient incident dialysis median hematocrit is 24.2% (interquartile range, 20.6%–27.5%). Given these facts, the selected population in our study could be the most representative of stage 5 CKD predialysis patients in Taiwan, who are likely to require hemodialysis eventually. Additionally, the study design calls for the analysis of drug interference of renal protection by analyzing the ACEI/ARB use within the 90 days after the first index day.

The confirmation of medication prescription (folic acid: 5 mg/tab; pentoxifyllinee: 400 mg/tab) was defined as >30 DDD (defined daily dose) before the second index date. The average daily maintenance dose for an average-size adult weighing 70 kg for the primary indication is the definition of the defined daily dose (DDD). The DDD is a standardized unit that provides a fixed reference that allows researchers to assess trends in drug consumption and comparison. The study population was categorized into four groups based on medication prescriptions for (1) folic acid, (2) pentoxifylline, (3) combination therapy, and (4) no medication. It is hard to distinguish whether the combination therapy group took folic acid and pentoxifylline at the same time or one after the other. However, we can confirm that both were provided in the combination therapy group.

This study was approved by the Institutional Review Board (IRB) at Taichung Tzu Chi Hospital (IRB: REC104-32). The IRB agreed that the inform consent of the patient could be waived because of the original identification numbers in NHIRD have been encrypted to protect patients’ privacy.

### 2.3. Statistical Analysis

The qualitative variables are presented as numbers and percentages, and the quantitative variables are expressed as means and standard deviations (SDs). The Chi-square test and Student *t*-test were used to examine the distributions of categorical variables and differences of continuous variables, respectively. The analysis of variance (ANOVA) and the Kruskal-Wallis test were applied for comparing two or more independent samples of different sample sizes. A *p*-value of <0.05 was considered a statistically significant difference. A histogram of predialysis duration was plotted to show the variability. The Kaplan–Meier method was used to determine dialysis-free survival between the different therapeutic groups. The log-rank test was introduced for statistical verification of survival curve differences. The multiple regression analysis was used to present the y-intercept (β regression coefficient) in different groups. SAS for Windows (version 9.2, SAS Institute Inc., Cary, NC, USA) and SPSS (version 19.0, SPSS Inc., Chicago, IL, USA) were used for the statistical analyses.

Since our study was extracted from the patient files in the Registry for Catastrophic Illness, all patients were confirmed entering the chronic dialysis status. We aimed to compare the duration between index dates. Which means that all cases have events (event status = 1). Everyone in our study had an event, and everyone in the study was observed a time to live. The nonparametric method (e.g., KM) can preliminarily estimate the survival equation and the cumulative risk equation without making any assumptions about the parametric assumption of survival time. Therefore, we tried to set a period of time about six months from the index date for KM survival curve plotting. Under the setting, the difference of each treatment group was extreme obvious presenting in dialysis-free survival. That’s why we finally tried to use the Kaplan–Meier method as presentation.

In practice, the distribution of most of the survival time may not meet the Weibull distribution. In other words, when it is impossible to determine the distribution of the survival time, it is not appropriate to use the parameter method analysis. At this point, consider abandoning the assumption of the distribution of survival time, using the nonparametric method (e.g., Kaplan–Meier) was another option.

For comparing the period of time in different groups, using ANOVA was sufficient for a significant difference. Furthermore, we want to understand the effect of different variables on extending time, thus the multiple regression analysis was chosen.

## 3. Results

We identified 103,646 patients 20–90 years old with an ESRD certification between January 2005 and December 2010. After excluding the patients with dialysis initiation before the ESA prescription, renal transplantation, and peritoneal dialysis, patients without the ACEI/ARB prescription, and patients who died within 90 days of the first index date, patients with cancer, and those with missing data, the remaining 10,954 individuals were enrolled in the study group. The patients were grouped according to the medication prescribed (Figure 1). Group A (*n* = 5073) received folic acid, group B (*n* = 1119) received pentoxifylline, group C (*n* = 2184) received both folic acid and pentoxifylline, and group D (*n* = 2578) received neither of those medications.

The mean age (±SD) was 60 ± 13 years in group A, 61 ± 12 years in group B, 61 ± 12 years in group C, and 60 ± 13 years in group D. Group A had a lower Charlson Comorbidity index (CCI) (1.7 ± 1.6), and women predominated (51.2%) compared with the other groups (Table 1).

Group C (folic acid + pentoxifylline) had the longest interval before the initiation of dialysis, with a mean of 244 ± 286 days (mean ± SD) and a median 145 days (25th–75th percentile: 58–327). In group A (folic acid), this interval was a mean of 216 ± 270 days and a median of 111 days (25th–75th percentile: 42–284), and for group B (pentoxifylline), the mean was 188 ± 229 days, and the median was 102 days (25th–75th percentile: 45–240). In group D (no folic acid or pentoxifylline), the mean duration was shortest, 166 ± 209 days, and the median was 89 days (25th–75th percentile: 36–217) from the start of the ESA treatment to the start of the maintenance dialysis (Table 2).

The times from the first index date to the second index date for the groups are plotted in Figure 2A. Further evaluation of the outcomes of patients with and without diabetes is presented in Table 3. Figure 2B,C show the different effects of medication on the duration from the ESA start to the initiation of chronic hemodialysis for patients with and without diabetes. The delay of dialysis for patients with diabetes was shorter than that of patients without diabetes. The Kaplan–Meier analysis with the log-rank test, which presents dialysis-free survival within six months, is shown in Figure 3. Folic acid and pentoxifylline, separately and together, delayed the need for dialysis.

The multiple regression analysis demonstrated that group C (both folic acid and pentoxifylline) had a y-intercept value of β = 84.52 (SE 7.39), whereas group A (folic acid) had a value of β = 41.55 (SE 6.11), and group B (pentoxifylline) had a value of β = 34.47 (SE 9.08). Considering group D (no folic acid or pentoxifylline) as the reference, group C (combination treatment) was superior to the other treatment groups. Even in patients with diabetes, the difference was statistically significant (Table 4).

## 4. Discussion

The purpose of this study was to investigate the efficacy of folic acid and pentoxifylline to delay the need for dialysis in patients with advanced CKD and to appraise the value and effectiveness of the treatment for delaying the need for dialysis in Taiwan. We observed that the combination of folic acid and pentoxifylline therapy resulted in the longest duration (median, 145 days) before the dialysis initiation in patients with advanced CKD, followed by folic acid (median 111 days), and pentoxifylline (median 102 days). These results were statistically significant in the multiple regression analyses.

Patients with CKD on potassium-restricted diets may have folic acid deficiencies [24]. We have reviewed the previous large folic acid study involving renal protection in a population of patients with CKD. ASFAST (Atherosclerosis and Folic Acid Supplementation Trial) was a multicenter, double-blinded, randomized controlled trial of high-dose folic acid (15 mg/d) therapy in 315 patients with stages 4 and 5 CKD [5]. The results showed that high-dose folic acid did not improve cardiovascular morbidity or mortality in patients with chronic renal failure. In addition, the HOST (Homocysteinemia in Kidney and End-stage Renal Disease) study was a prospective, double-blind, randomized controlled trial that enrolled patients with stage 4 and 5 CKD and patients with ESRD with hyperhomocysteinemia [25]. With 1003 patients in each arm, it is the largest trial to date of high-dose folic acid (40 mg/d) and B vitamin therapy, which did not improve survival or delay the time to initiating dialysis in the study patients. Additionally, the DIVINe (Diabetic Intervention with Vitamins to Improve Nephropathy) trial disclosed that the treatment with folic acid (2.5 mg/d) and B vitamins resulted in a greater decrease in GFR and an increase in vascular events in 238 patients with diabetic nephropathy [10]. However, these trials were conducted in grain fortification regions. On the contrary, only the Renal Substudy of the China Stroke Primary Prevention Trial (CSPPT) was undertaken in a population without folic acid fortification. The supplementation of folic acid significantly delayed the progression of CKD among patients with mild-to-moderate CKD. Low-dose folic acid (0.8 mg/d) without B vitamins was given in the CSPPT, and the benefits between this and baseline results in the other studies were clearly illustrated. The discrepancy may be partly explained by differences in treatment schemes (low-dose folic acid) and patient characteristics (no folic acid fortification area). In our study, we applied the CSPPT renal study assumptions and targeted predialysis duration in similar regional populations without folic acid fortification [26,27]. Similar to the fact that the mortality rate relates to cancer, the ESRD dialysis relates to renal death. Therefore, prolonging the survival period of renal life before dialysis (even dialysis-free time) is particularly meaningful.

Pentoxifylline is a nonselective phosphodiesterase inhibitor and is used clinically to treat peripheral vascular disorders because of its anti-inflammatory and immune-modulatory effects. The investigators of the HERO (Handling Erythropoietin Resistance with Oxpentifylline) trial, a randomized placebo-controlled trial, have announced that pentoxifylline increased hemoglobin concentration in anemic patients with CKD. Furthermore, when combined with renin-angiotensin system blocking agents, the investigators of the PREDIAN (Pentoxifylline for Renoprotection in Diabetic Nephropathy) trial reported it delayed the eGFR loss in stage 3–4 kidney disease in diabetic patients [28,29]. Additionally, a nationwide database analysis showed an added protective effect for pentoxifylline in advanced CKD treated with the renin-angiotensin-aldosterone system blockade [15]. However, clinical comparisons of the effects of folic acid and pentoxifylline for renal protection are rare. We sought to fill this gap in the literature and to clarify the vague therapeutic zone and confirm the treatment efficacy in Taiwan by analysis of the nation-wide database.

The results of our research are notable. First, our study is the first designed to compare the effects of folic acid and pentoxifylline on renal protection in predialysis patients with advanced CKD. Second, like the CSPPT renal substudy, the treatment results for low-dose folic acid were consistent in similar non-fortification regions. Furthermore, based on the previous comprehensive studies, we have focused on the renal protective effect of folic acid plus pentoxifylline and have extended our study groups from mild-to-moderate CKD to CKD stage 5 without dialysis. Third, based on the benefits of a nation-wide database, we had a large number of patients in each arm of the study.

Despite the power of our retrospective observational research, our study still has several limitations. First, the NHIRD lacked laboratory data for eGFR, without which it is impossible to judge whether there is any difference in the eGFR of the four groups. However, we could extrapolate on the basis of the strict NHI reimbursement regulations, since most clinical therapeutic decisions were made according to laboratory data, we could confirm the clinical decisions by investigating the procedure codes in the NHIRD database instead of laboratory data. Once the course of clinical events was complete, we assumed that it must correspond to the necessary laboratory data. Otherwise, regular health insurance audits would punish providers and potentially remove them from the reimbursement system; compliance with the regulations was high. Therefore, the selected target patients were validated under the same strict NHI reimbursement standards. Despite lacking laboratory data (such as creatinine and eGFR) in the NHIRD study, the results are still persuasive under the compensation scheme of the regulations.

Second, this study was not a randomized clinical trial in design. The retrospective nature of the study renders it vulnerable to the selection bias. Nonetheless, the retrospective observational design by nature reduced the potential risk of violating medical ethics. Large-scale, randomized controlled trials are impractical in Taiwan whereas well-designed observational studies can yield comparable results [30,31]. Although observational studies cannot define causation, seeking trends and exploring possible interventions to be studied in randomized controlled trials is a valuable advantage of a nation-wide database such as the NHIRD.

Third, the results of this study may not be generalizable to all patients with advanced CKD. Patients with advanced CKD but no ESA therapy for obvious renal anemia were excluded. Additionally, self-pay ESA was not recorded by the NHI. Patients with CKD stage 5 who had self-pay folic acid and pentoxifylline were not enrolled either. Hence, further research is warranted.

Fourth, the ESA therapy and medication use was based on prescriptions recorded in the NIH database. Thus, we did not have the opportunity to monitor the patients for the compliance. However, nearly all Taiwanese patients with a principal diagnosis of CKD have received a multidisciplinary care (MDC) program since 2004. With education and team care, pre-ESRD patients were found to have more effective medication prescription under multidisciplinary care intervention [32,33,34,35]. Therefore, we’d like to be persuaded that the compliance of the patient was high under the CKD care programs.

## 5. Conclusions

In advanced CKD patients with the ESA treatment, folic acid and pentoxifylline may delay the initiation of hemodialysis in the study patients. The combination therapy was more effective than either folic acid or pentoxifylline alone, even in patients with DM.

## Figures and Tables

**Figure 1 nutrients-11-02192-f001:**
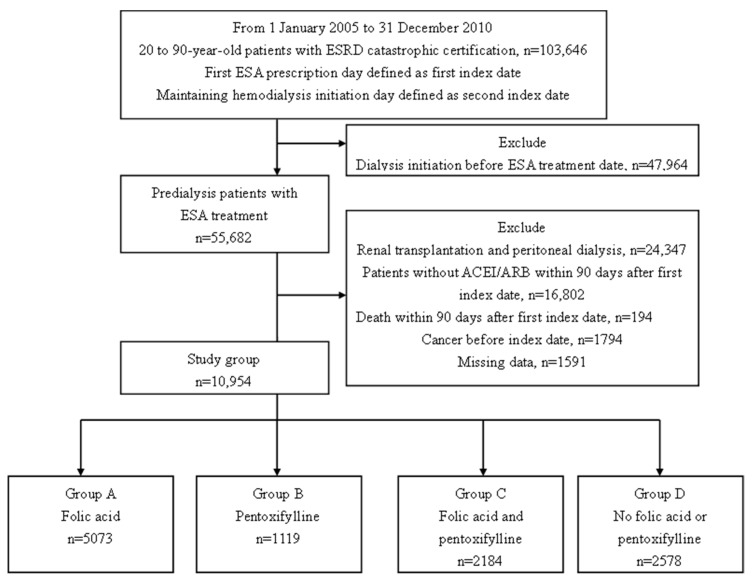
Patient selection flowchart. Abbreviations: ACEI, angiotensin-converting enzyme inhibitor; ARB, angiotensin II receptor blocker; ESRD, end-stage renal disease; ESA, erythropoiesis-stimulating agent.

**Figure 2 nutrients-11-02192-f002:**
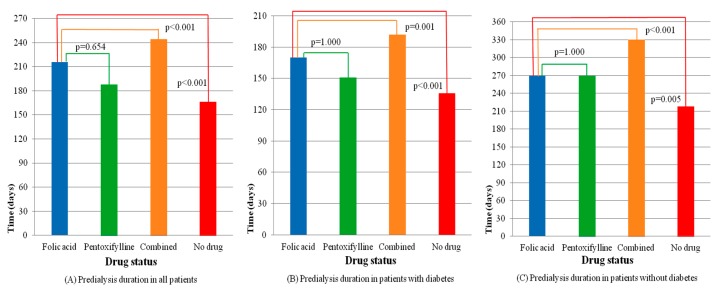
Comparisons of the predialysis duration among drug groups in patients with and without diabetes. (**A**) Predialysis duration in all patients, (**B**) Predialysis duration in patients with diabetes, (**C**) Predialysis duration in patients without diabetes.

**Figure 3 nutrients-11-02192-f003:**
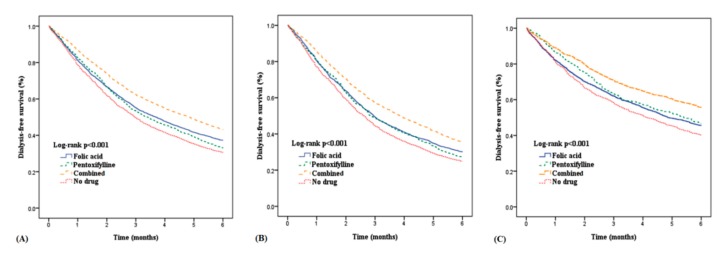
Dialysis-free survival in patients by drug groups among patients with and without diabetes. (**A**) Dialysis-free survival in all patients, (**B**) Dialysis-free survival in patients with diabetes, (**C**) Dialysis-free survival in patients without diabetes.

**Table 1 nutrients-11-02192-t001:** Baseline characteristics *.

Characteristic	Group A	Group B	Group C	Group D	*p*-Value
Patient No.	5073	1119	2184	2578	
Age, mean ± SD (year)	60 ± 13	61 ± 12	61 ± 12	60 ± 13	<0.001
CCIS, mean ± SD	1.7 ± 1.6	2.2 ± 1.6	2.0 ± 1.6	1.9 ± 1.6	<0.001
Sex					<0.001
Male	2476 (48.8)	667 (59.6)	1212 (55.5)	1368 (53.1)	
Female	2597 (51.2)	452 (40.4)	972 (44.5)	1210 (46.9)	
SES					0.060
Low	1711 (33.7)	382 (34.1)	763 (34.9)	949 (36.8)	
Moderate and high	3362 (66.3)	737 (65.9)	1421 (65.1)	1629 (63.2)	
Urbanization					<0.001
Non-urban	3558 (70.1)	891 (79.6)	1625 (74.4)	1938 (75.2)	
Urban	1515 (29.9)	228 (20.4)	559 (25.6)	640 (24.8)	
Geographic region					<0.001
Northern/central	2981 (58.8)	838 (74.9)	1648 (75.5)	1451 (56.3)	
Southern/eastern	2092 (41.2)	281 (25.1)	536 (24.5)	1127 (43.7)	

* Data are number (%) unless otherwise indicated. Group A: Folic acid; Group B: Pentoxifylline; Group C: Folic acid + pentoxifylline; Group D: No folic acid or pentoxifylline. Abbreviations: CCIS, Charlson Comorbidity index; SES, socioeconomic status.

**Table 2 nutrients-11-02192-t002:** Comparison of the predialysis duration between different treatment groups.

Characteristic	Group A	Group B	Group C	Group D	*p*-Value
Patient No.	5073	1119	2184	2578	
Duration (days)					
Mean ± SD	216 ± 270	188 ± 229	244 ± 286	166 ± 209	<0.001 ^+^
Median	111	102	145	89	<0.001 ^#^
25th pctl–75th pctl	42–284	45–240	58–327	36–217	

^+^ ANOVA; ^#^ Kruskal-Wallis test. Group A: Folic acid; Group B: Pentoxifylline; Group C: Folic acid + pentoxifylline; Group D: No folic acid or pentoxifylline. Abbreviations: ANOVA, analysis of varaiance; pctl, percentile.

**Table 3 nutrients-11-02192-t003:** Comparison of the predialysis duration among patients with and without diabetes by treatment group.

Characteristic	Diabetes					No diabetes				
	Group A	Group B	Group C	Group D	*p*-Value	Group A	Group B	Group C	Group D	*p*-Value
Patient No.	2733	774	1355	1614		2340	345	829	964	
Duration, mean ± SD (days)	170 ± 217	151 ± 181	192 ± 229	136 ± 171	<0.001 ^+^	269 ± 312	269 ± 296	330 ± 343	218 ± 252	<0.001 ^+^
Median (days)	89	86	116	79	<0.001 ^#^	146	162	220	124	<0.001 ^#^
25th–75th pctl	39–221	40–197	50–260	33–178		47–384	61–373	74–476	42–299	

^+^ ANOVA; ^#^ Kruskal-Wallis test Group A: Folic acid; Group B: Pentoxifylline; Group C: Folic acid + pentoxifylline; Group D: No folic acid or pentoxifylline. Abbreviations: ANOVA, analysis of variance; pctl, percentile.

**Table 4 nutrients-11-02192-t004:** Multiple regression analysis.

Characteristic	Total			Diabetes			No Diabetes		
	β	SE	*p*-Value	β	SE	*p*-Value	β	SE	*p*-Value
Intercept	180.36	8.96	<0.001	150.07	9.95	<0.001	158.28	16.48	<0.001
Drug status *									
No drug (ref.)									
Pentoxifylline	34.47	9.08	<0.001	17.40	9.02	0.054	72.48	19.04	<0.001
Folic acid	41.55	6.11	<0.001	33.27	6.45	<0.001	47.00	11.55	<0.001
Combined	84.52	7.39	<0.001	57.50	7.62	<0.001	125.63	14.40	<0.001
Sex									
Male (ref.)									
Female	34.81	4.85	<0.001	9.81	5.15	0.057	64.76	9.05	<0.001
SES									
Low (ref.)									
Moderate and high	14.62	5.13	0.004	3.67	5.45	0.501	30.80	9.59	0.001
Urbanization									
Non-urban (ref.)									
Urban	7.12	5.56	0.201	7.88	5.89	0.181	7.48	10.41	0.473
Geographic region									
Northern/central (ref.)									
Southern/eastern	23.06	5.13	<0.001	1.81	5.44	0.739	56.03	9.58	<0.001
CCIS	−27.88	1.53	<0.001	−10.42	1.75	<0.001	−25.30	4.15	<0.001

Abbreviations: SE, standard error; CCIS, Charlson Comorbidity index; ref., reference; SES, socioeconomic status. * The analyses of drug status were adjusted for sex, SES, urbanization, geographic region, and CCIS.

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
