# Peer review of "Comparing the Effect of Folic Acid and Pentoxifylline on Delaying Dialysis Initiation in Patients with Advanced Chronic Kidney Disease"

_nutrients, 2019, doi:10.3390/nu11092192_

Round 1

Reviewer 1 Report

Comparing the Effect of Folic Acid and Pentoxifylline on Delaying Dialysis Initiation in Patients with Advanced Chronic Kidney Disease.

This study compared the effects of folic acid and pentoxifylline for delaying the initiation of dialysis. This study is well written and concise there is no real hypothesis as to why it was done.  While there is rationale for use of folic acid and pentoxifylline to delay the progression of CKD, why were these two meds chosen to study together?  Why did you exclude patients on an ACE or ARB?  Why did you include socioeconomic status and geographic region in your analysis?

Strengths:

·         We need investigation on ways to slow the progression of CKD and this is an attempt

·         Large cohort of patients

·         Statistics and KM curve are clear

 Weakness:

·         Retrospective

·         No hypothesis as to why the two meds were chosen to study together

 Recommendations:

·         Even though this study is retrospective, format a hypothesis

·         Include other medications that may affect the progression of CKD such as: ACE, ARB, ASA, Plavix, ASA and Plavix, Coumadin, Eliquis and other NWAC

·         Include other demographics and PMH that may affect progression of CKD such as: BMI, age hypertension, CAD, PVD, CVA, CHF, history of thrombosis

Reviewer 2 Report

Dr Yang and his team have conducted an excellent retrospective study that explores a critical need in the management of CKD, which is how to delay progression to hemodialysis. Their findings are highly relevant, particularly in populations without standard food fortification with folic acid. 

Here the authors explore the role of folic acid, pentoxifylline, a combination of the two, and no therapy. Their findings suggest a beneficial role of either therapy compared to no therapy, but their significant finding is the uniquely protective role of combination therapy.

I recommend acceptance with at most minimal grammatical revisions as would be suggested by the copy-editor.

Author Response

Dear editor and reviewer:

    Thank you very much for your valuable suggestions. We deeply appreciate your acceptance. 

Reviewer 3 Report

This is an observational retrospective study. All the causal wording present in the manuscript (e.g. “Folic acid and pentoxifylline delay dialysis” in the running title) should be changed with more cautious wording There is the potential for significant confounding by a number of sources, including renal function at index date (the authors do not have information on actual eGFR and thus cannot adjust the analysis for such variable) and confounding by indication. In fact, the authors report only results from unadjusted analyses that are not very helpful in the setting of an observational study Graphs are difficult to read Rather than using the ANOVA to compare time to an event, the authors should have used a parametric survival model (e.g. Weibull)

Round 2

Reviewer 1 Report

OK to accept with revisions 

Author Response

Thank you very much for your valuable suggestions. We deeply appreciate your acceptance.